# NFL-BA: Near-Field Light Bundle Adjustment for SLAM in Dynamic Lighting

**Andrea Dunn Beltran**[*1]**, Daniel Rho**[*1]**, Marc Niethammer**[2]**, Roni Sengupta** [1]

[1] University of North Carolina at Chapel Hill    [2] University of California San Diego

[*] Equal contribution

{asdunnbe, dnl03c1, ronisen}@cs.unc.edu, mniethammer@ucsd.edu

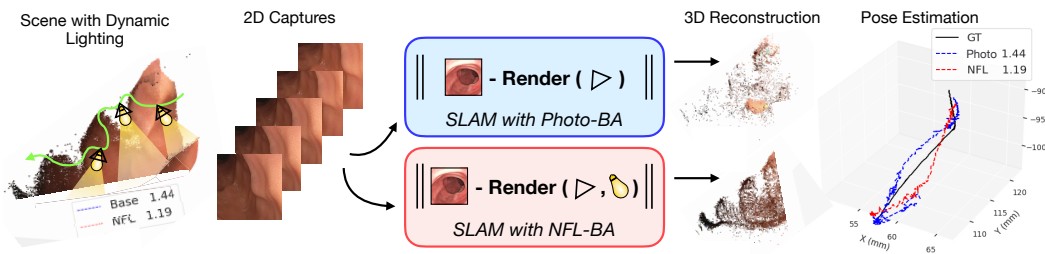

Figure 1: **NFL-BA enhances tracking and mapping in neural rendering-based SLAM** (e.g., MonoGS [27]) by explicitly modeling dynamic near-field lighting, with applications in endoscopy.

## Abstract

Simultaneous Localization and Mapping (SLAM) systems typically assume static, distant illumination; however, many real-world scenarios, such as endoscopy, subterranean robotics, and search & rescue in collapsed environments, require agents to operate with a co-located light and camera in the absence of external lighting. In such cases, dynamic near-field lighting introduces strong, view-dependent shading that significantly degrades SLAM performance. We introduce Near-Field Lighting Bundle Adjustment Loss (NFL-BA) which explicitly models near-field lighting as a part of Bundle Adjustment loss and enables better performance for scenes captured with dynamic lighting. NFL-BA can be integrated into neural rendering-based SLAM systems with implicit or explicit scene representations. Our evaluations mainly focus on endoscopy procedure where SLAM can enable autonomous navigation, guidance to unsurveyed regions, blindspot detections, and 3D visualizations, which can significantly improve patient outcomes and endoscopy experience for both physicians and patients. Replacing Photometric Bundle Adjustment loss of SLAM systems with NFL-BA leads to significant improvement in camera tracking, 37% for MonoGS and 14% for EndoGS, and leads to state-of-the-art camera tracking and mapping performance on the C3VD colonoscopy dataset. Further evaluation on indoor scenes captured with phone camera with flashlight turned on, also demonstrate significant improvement in SLAM performance due to NFL-BA.

## 1   Introduction

Simultaneous Localization and Mapping (SLAM) enables autonomous agents to build a spatial map of an unknown environment while estimating their own poses within it, with wide-ranging applications in robotics, computer vision, autonomous vehicles, and scientific imaging. Most SLAM systems [38, 34, 7, 58, 60, 56, 49, 20, 27, 16] assume an autonomous agent navigating an environment with distant, static illumination, e.g., a self-driving car in the streets, and they optimize a Photometric Bundle Adjustment loss where they minimize an error between the captured image and the re-rendered image using estimated 3D scene and camera poses.

39th Conference on Neural Information Processing Systems (NeurIPS 2025).

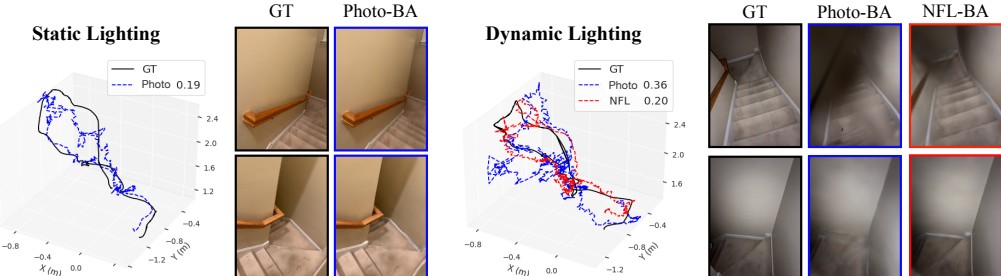

Figure 2: **MonoGS performance under (1) distant static lighting and (2) dynamic near-field lighting from a co-located flashlight.** Standard photometric BA performs well under static lighting but fails under dynamic lighting, degrading both trajectory and map quality. NFL-BA restores performance under dynamic lighting, matching the quality of the static-light setup.

However, many scientific and safety-critical applications demand that autonomous agents operate in environments devoid of external illumination, relying instead on self-mounted light sources. For example, in endoscopy procedures, a slender flexible tube with a co-located light and camera is used to inspect internal organs such as the airway and the colon [9, 45, 33, 23, 51, 17]. Accurate trajectory estimation is crucial for reliably guiding instruments to areas of interest, mapping anomalies, and avoiding tissue damage during navigation. In subterranean search-and-rescue or collapsed-building inspection, robots rely on onboard lamps to explore unstable voids; slight errors in pose estimation can accumulate into large drift, leading to misaligned maps, missed victims, or costly back-tracking.

Despite the prevalence of these use cases, current SLAM systems perform poorly under such conditions (see Fig. 2). This performance drop is primarily due to the effects of *dynamic near-field lighting*, where the only illumination is co-located with the camera and moves with it. Dynamic near-field lighting causes different points of the surface to receive different intensities of light at each time step, depending on the distance and orientation of the point to the camera, introducing strong, view-dependent shading. These lighting artifacts significantly impair both feature-based and direct (photometric) tracking, resulting in substantial failures in mapping accuracy and pose estimation.

To alleviate these issues, we propose a new Bundle Adjustment loss that accounts for dynamic near-field lighting. Our key intuition is that the shading effect of the captured image can provide valuable information about the relative distance and orientation between the surface and the camera. With this, we formulate a Near-Field Lighting Bundle Adjustment loss, NFL-BA, where we optimize the surface geometry and the camera parameters such that the rendered image has shading variations that match the relative distance and orientation between the surface and the camera. Our NFL-BA loss can be applied to any neural rendering-based SLAM algorithm, i.e., with neural implicit and explicit 3D Gaussian scene representation.

In this paper, we specifically focus on demonstrating how NFL-BA can improve the performance of existing SLAM systems for 3D reconstruction and localization from endoscopy videos. SLAM can enable autonomous navigation through internal organs and guide physicians to unsurveyed regions to improve physicians' situational awareness by providing 3D visualizations, and can help measure organ shapes. We evaluated NFL-BA with two state-of-the-art 3DGS-based SLAM systems, general-purpose MonoGS [27] and endoscopy-specific EndoGSLAM [45], and one neural implicit SLAM, NICE-SLAM [58], by replacing their Photometric Bundle Adjustment loss with NFL-BA loss. We observe that the NFL-BA loss improves the performance of all SLAM algorithms on average when using ground-truth or estimated depth maps on the C3VD colonoscopy dataset. For example, NFL-BA significantly improves MonoGS by reducing camera tracking error by 37% (3.48 to 2.18 mm) and camera mapping error by 38% (1.59 to 0.99 mm) when initialized by PPSNet depth[35].

Additionally, we also demonstrate the effectiveness of NFL-BA on indoor rooms captured with a moving co-located light and camera without any external light source, mimicking agent navigation during search & rescue and covert military operations. By replacing incorporating our NFL-BA loss, we see an average improvement of ∼35% in pose estimation across all scenes.

## 2   Related Works

**Dense SLAM and Bundle Adjustment.** Early SLAM pipelines focused on sparse feature matching for pose estimation and mapping [30, 6, 42, 11]. With advancements in neural scene representations

several proposed SLAM frameworks [58, 2] generate dense, pixel-level that yield more detailed and robust reconstructions. More recently, 3D Gaussian surface methods have demonstrated real-time rendering with high-fidelity mapping[20, 27, 49, 16, 10, 53].

These dense SLAM approaches all rely on a core Bundle Adjustment step. Bundle Adjustment (BA) alternatively optimizes camera parameters and surface geometry by minimizing errors across multiple frames. Traditional geometric BA aligns detected 2D feature points to their 3D counterparts by minimizing reprojection error, assuming static lighting and Lambertian surfaces [14]. Although effective in controlled environments, it struggles in complex or low-texture scenes. Photometric BA (Photo-BA) [1] incorporates pixel intensities into the optimization process, minimizing photometric re-projection errors and proving advantageous in environments where feature matching fails [11]. However, Photo-BA does not exploit the correspondence cues provided by dynamic or near-field lighting where image intensities vary across frames.

**Near-field Lighting models.** Near-field lighting has been leveraged for 3D reconstruction tasks like monocular depth and surface normal estimation [35, 57] and Photometric Stereo [21]. Some of these approaches [35, 21] use a near-field lighting representation as input to a CNN along with captured images for predicting surface normal and geometry. In the context of Endoscopy, LightDepth [37] and PPSNet [35] demonstrated the effectiveness of near-field lighting to enhance depth estimation. LightNeus [3] exploited the inverse-square law for light decay to improve endoscopic surface reconstruction, however with known camera parameters and pre-operative 3D CT scan.

*It has never, however, been used for Simultaneous Localization & Mapping (SLAM) problems, let alone in combination with neural rendering methods.* To this end, we propose a Bundle Adjustment Loss with Near-Field Lighting (NFL-BA), considering the most commonly available single co-located camera & light in the endoscope or other autonomous agents.

**Dynamic Lighting in SLAM.** Visual SLAM performance often degrades under illumination changes such as exposure shifts, specularities, and varying color temperature. Early photometric calibration methods jointly optimize camera intrinsics, exposure, and scene depths to normalize brightness variations in real time [11] while probabilistic SLAMs with unscented filtering further stabilizes pose estimates under uncertain lighting conditions [26]. More recently, learning-based matchers [22, 48] adapt descriptors to cope with complex lighting variations. None of these methods, however, explicitly model near-field lighting geometry to handle this co-located light setting.

**SLAM in endoscopy.** Early works [40, 12] demonstrated the feasibility of applying SLAM in such environments by addressing dynamic lighting and tissue deformation. Researchers have often used a mixture of supervised learning on synthetic and self-supervised learning on real endoscopy datasets for tailoring SLAM frameworks to endoscopy with complex camera motion [25, 55, 46] and developed novel endoscopy SLAM frameworks [36, 29, 18]. However these techniques often struggle with challenging sequences from both synthetic and clinical data. Recently, neural rendering-based methods [39, 23, 45, 51, 13, 15] have proved especially effective in generating high-quality details and modeling textureless regions with a large number of Gaussians. In this work, we adopt neural rendering approaches and explicitly model the near-field lighting effects, alleviating dynamic lighting challenges and improving performance.

## 3  Background

In this section, we review the general framework of neural rendering-based SLAM. We represent the camera at time $t$ by its extrinsics $P_t = [R_t, T_t] \in \mathbf{SE}(3)$ and known intrinsics $K$, yielding the projection $\pi_t = K P_t$. We assume the camera intrinsic $K$ to be the same for all frames and known or calibrated ahead of time. Pixels are denoted $p$ and 3D camera-space points by $x$.

In neural rendering, scene parameters $\Theta$, whether in the form of neural networks or primitives, encode visual and geometric information, such as colors $c_i$ and occupancy $\alpha_i$. Given $\Theta$ and $P_t$, we can get the color $\hat{C}(\cdot)$ and the depth $\hat{D}(\cdot)$ of a pixel $p$ from a frame at time $t$ as follows [28, 20]:

$$\hat{C}(p) = \sum_{i \in \mathcal{N}} c_i \alpha_i \Pi_{j=1}^{i-1} (1 - \alpha_j), \quad \hat{D}(p) = \sum_{i \in \mathcal{N}} z_i \alpha_i \Pi_{j=1}^{i-1} (1 - \alpha_j) \qquad (1)$$

where $\mathcal{N}$ denotes the group of samples for a pixel $p$, with $\alpha_i$ representing the occupancy of the $i$-th sample, and $z_i$ denotes its distance from the camera center.

To optimize $P_t$ and $\Theta$, dense SLAM methods typically use rendering loss $\mathcal{L}_{ren}$, reducing the rendering errors between the rendered and captured images [58, 27, 49] and, if estimated or ground

truth depth maps are available, an additional depth loss $\mathcal{L}_{geo}$ can be added [43]. Typically, these losses take the form of $L^p$ norm as follows with variations with $M_t$ as a pixel-wise mask:

$$\mathcal{L}_{ren} = \|M_t \odot (\hat{C} - C)\|_p, \quad \mathcal{L}_{geo} = \|M_t \odot (\hat{D} - D)\|_p \tag{2}$$

Bundle adjustment optimizes both $P_t$ and $\Theta$ using the following combined loss:

$$\text{Photo-BA:} \quad \min \sum_{t \in \mathcal{W}} \lambda_{ren}\mathcal{L}_{ren}(\hat{C}, C; M_t) + \lambda_{geo}\mathcal{L}_{geo}(\hat{D}, D; M_t) \tag{3}$$

where $\mathcal{W}$ denotes the set of frames used for the bundle adjustment and the hyperparameters $\lambda_{ren}$ and $\lambda_{geo}$ are the loss weights. Additionally, the objective function can include any other regularization terms, such as artifact suppressing [27] or opacity regularization [59].

During the Mapping stage, both $\Theta$ and $P_t$ are optimized over a set of keyframes. The exact algorithm for keyframe selection, keyframe update and optimization strategies for tracking and mapping phase vary between different SLAM approaches and their specific objectives.

**Implicit Neural Representations.** Neural field-based SLAM methods [41, 58, 44, 60, 38] uses a set of neural networks $F(x, d; \Theta) \rightarrow (c_i, \sigma_i)$, optimized to estimate the color $c_i$ and the volume density $\sigma_i$ for an input 3D coordinate $x$ and the view direction $d$. The the occupancy can be calculated from the volume density $\sigma_i$ and the distance between adjacent samples $\delta_i$ as $\alpha_i = 1 - \exp(-\sigma_i\delta_i)$.

**3D Gaussian Splatting.** For 3D Gaussian Splatting [20] SLAM methods, the scene is represented by a set of Gaussians with mean $\mu^i$, covariance $\Sigma^i$ in world space, color $c_i$, and opacity $\alpha^i$. The shape parameters and occupancy $\alpha^i$ of the *splatted* 2D Gaussians are computed as follows:

$$\bar{\mu}_t^i = \pi_t\mu^i, \quad \bar{\Sigma}_t^i = J_t R_t \Sigma^i R_t^T J_t^T, \quad \alpha_i = \alpha^i \exp(-\frac{1}{2}(p - \bar{\mu}_t^i)^\top (\bar{\Sigma}_t^i)^{-1} (p - \bar{\mu}_t^i)) \tag{4}$$

where $J_t$ is the Jacobian of the projection $\pi_t$, $p$ denotes a pixel coordinate, and $\bar{\mu}_t^i, \bar{\Sigma}_t^i$ are the splatted mean and covariance of Gaussian $\mathcal{G}^i$ in pixel space.

## 4 Near-Field Light Bundle Adjustment

We introduce a novel Near-Field Lighting based Bundle Adjustment loss, NFL-BA, that integrates near-field lighting with neural-rendering 3D scene representations to improve performance of existing SLAM systems on images captured with dynamic lighting co-located with the camera. Our proposed NFL-BA can replace commonly used Photometric Bundle adjustment loss, defined in Eq. 3, within neural-rendering based SLAM framework. Photo-BA typically optimizes scene appearance parameter as RGB color, which is suffient when the illumination on

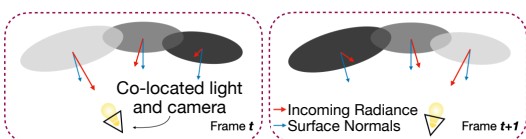

Figure 3: **Illustration of our key idea.** As the co-located light and camera, moves through the scene, different 3D Gaussians on the surface receive different intensities of light (red arrow), dependent on the relative distance and orientation between the 3D Gaussian and the camera.

each scene point remains the constant throughout the capture. However, for scenes with a dynamic light co-located with a moving camera, the illumination received at each point varies per frame as the camera and the light moves through the scene. In this setting, the illumination received at each point depends on the relative distance and orientation between the point and the camera, as conceptualized in Fig. 3. Thus continuing to model scene appearance as simple RGB color is inaccurate for dynamic near-field lighting as it doesn't separate effects of illumination due to camera movement from the intrinsic view-independent color of the scene, i.e. albedo.

Our goal is to explicitly model surface appearance as albedo and separate near-field lighting effects from it. To accurately model dynamic lighting we then represent near-field illumination effects with camera pose and scene geometry. In sec. 4.1 we describe our image formation model using neural rendering framework that will decompose the surface appearance into albedo and incoming lighting, which will be further represented as a function of scene geometry and camera pose. Then in sec. 4.2, we will use this image formation to create the Near-Field Bundle Adjustment loss and show how it can be easily integrated into neural rendering based SLAM framework.

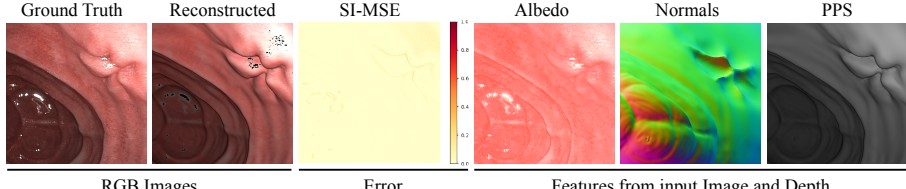

| Ground Truth | Reconstructed | SI-MSE | Albedo | Normals | PPS |

| RGB Images | Error | Features from input Image and Depth |

Figure 4: **Image Formation Validation.** We show that C3VD images captured with a real endoscope conform to our co-located light-camera and zero attenuation $\beta$ image formation model, as indicated by very low per-pixel scale-invariant MSE between the original image and the reconstructed image with masked-out specular regions.

## 4.1 Image Formation with Near-Field Lighting

We consider an image-formation model under near-field lighting for a single image following previous works [19, 35]. Each pixel $p$ and the corresponding three-dimensional point $x_p$ in the camera space receives different light intensities and directions, characterized by the light source to surface direction $L^d(\cdot)$ and attenuation term $L^a(\cdot)$, as follows:

$$L^d(x_p) = \frac{x_p - x_L}{\|x_p - x_L\|}, \quad L^a(x_p) = \frac{(L^d(x_p)^\top f)^\beta}{\|x_p - x_L\|^2}, \tag{5}$$

where $x_L$ is the location of the light source, $f$ is the forward (optical axis) vector. $\beta$ is an angular attenuation coefficient, and will be discussed in sec. 4.2.

Assuming a diffuse reflectance model, which has proven effective for depth estimation in endoscopic scenes [35], we can approximate the rendered image at each pixel $\hat{C}(\cdot)$ as:

$$PPS(x_p) = L^a(x_p) \cdot (L^d(x_p)^\top n(x_p)), \qquad \hat{C}(p) = \rho(x_p)PPS(x_p), \tag{6}$$

where $\rho(\cdot)$ and $n(\cdot)$ are albedo and normal at position $x_p$ of pixel $p$ respectively. $PPS(\cdot)$ is a per-pixel shading term. Note that existing approaches that uses this near-field light image formation model [19, 35] uses pixel-based representation to predict depth map or surface geometry from images captured from a single viewpoint only. In this paper, we extend the Near-Field Image Formation model beyond single-view pixel-based representation to multi-view 3D representation.

Our key insight is that the standard volumetric rendering equation can be modified to incorporate the near-field lighting model described in eq. 6, while keeping the overall SLAM pipeline intact. In our framework, we reinterpret the direct color ($c_i$ in eq. 3) as the product of the albedo $\rho(\cdot)$ and the shading term $PPS(\cdot)$, which models dynamic near-field lighting. Note that both albedo and shading is defined directly on the 3D neural representations, i.e. neural radiance field or 3D gaussians, and not in pixel-space. This leads to the modified rendering equation under near-field lighting:

$$\hat{C}_{pps}(p) = \sum_{i \in \mathcal{N}} \rho(x_i)PPS(x_i)\alpha_i\Pi_{j=1}^{i-1}(1 - \alpha_j) \tag{7}$$

Note that eq. 6 represents a special case of eq. 7 where a single sample is considered and the occupancy $\alpha_i$ equals one. Our image formation model assumes diffuse reflectance and no angular attenuation, to reduce the complexity of the modeling. While it is easy to extend the image formation model to handle specular reflectance and angular attenutation of lighting, this leads to additional parameters that needs to be optimized during the Bundle Adjustment.

**Angular attenuation.** Following previous works [?, 35], we simplify the near-field light image formation model by setting the attenuation coefficient $\beta$ in eq. 5 to zero. This effectively ignores the directional fall-off component, reducing the light attenuation term to a simple inverse-square fall-off $L^a(x_p) = 1/\|x_p - x_L\|^2$. This simplification is justified because the angular attenuation in settings like endoscopy is often negligible compared to the inverse square law attenuation, and estimating $\beta$ accurately can be challenging due to variations in endoscope designs. In future work, we plan determine the optimal value of $\beta$ for different systems and incorporate the light direction vector $r_t^e$ for more accurate modeling which can further improve camera rotation during Bundle Adjustment.

**Empirical validation of our image-formation model on colonoscopy image.** Fig. 4 provides an example colonoscopy image from the C3VD dataset [4], showing the accuracy of the near-light field model (Eq. 6) with $\beta$ of 0. Albedo was estimated by converting each RGB image to HSV color

space, setting the value channel to 1 across all pixels, then converting the modified image back to RGB space. This standardizes pixel intensity variations, approximating a reflectance map where illumination effects are minimized, but does not strictly represent ground truth albedo. As shown, the image formulation model is sufficient to represent endoscopic scenes with low reconstruction errors.

## 4.2 Near-Field Light Bundle Adjustment Loss

Next, we will re-define the Photometric Bundle Adjustment loss of eq. 3 using the near-field lighting based image formation model defined in eq . 7 expressed as follows:

$$\text{NFL-BA:} \quad \min \sum_{t \in \mathcal{W}} \lambda_{ren}\mathcal{L}_{ren}(\hat{C}_{pps}, C; M_t) + \lambda_{geo}\mathcal{L}_{geo}(\hat{D}, D; M_t) \tag{8}$$

where $\hat{C}_{pps}$ denotes the rendered image with near-field lighting-incorporated volumetric rendering equation (Eq. 7). This reformulation seamlessly integrates near-field lighting cues into the neural rendering framework without altering the rest of the SLAM framework. Since our formulation is confined solely to the rendering process, and thus to the bundle adjustment, we do not modify or replace any other SLAM components for fair comparison. This design choice enables easier integration with existing neural rendering-based SLAM methods.

**Choice of image space in optimization.** Note that many settings, and especially endoscopy, frames are stored in standard sRGB color space, whereas our near-field shading term $PPS(\cdot)$ is computed in a linear space. To ensure consistency, we apply an inverse gamma correction of $\gamma = 2.2$ to the sRGB images before computing $PPS$, or equivalently, gamma-correct the linear PPS output by $\gamma = 1/2.2$ when rendering back to sRGB. This step aligns the lighting model with the true photometric intensities and prevents bias from the nonlinear sRGB transfer function.

**Normal calculation during Bundle Adjustment.** To calculate the normals $n(\cdot)$ from neural fields, we utilize the direction of the gradient of the occupancy with respect to the spatial coordinates as follows [5]: $n(x_i) = -\nabla\sigma(x_i)/\|\nabla\sigma(x_i)\|$. For Gaussian Splatting, we use the shortest axis of each Gaussian as its normal, following [52, 8, 47]. In both cases, we ensure the computed normal is oriented towards the camera by enforcing $n(x)^\top L^d(x)$ to be positive. Otherwise, we flip the normals by multiplying them by -1 for stability.

## 5 Evaluation

Our proposed method is a plug-in approach that can be applied to any existing neural-rendering-based SLAM framework. We first test our method on endoscopy videos using one neural implicit SLAM, NICE-SLAM [58], as well as two existing 3DGS-SLAM frameworks: the general-purpose MonoGS [27] and the endoscopy-specific EndoGSLAM [45]. In each case, we replace the standard Photometric Bundle Adjustment loss (3) with our proposed equation NFL-BA loss (8). Additionally, we also test MonoGS [27] on self-captured indoor scenes with a co-located light and camera.

Table 1: **Quantitative Evaluation on the C3VD [4]** dataset with oracle depth map. Replacing Photometric BA with NFL-BA significantly improves tracking quality of two state-of-the-art 3D Gaussian SLAMs, MonoGS [27] and EndoGS [59], and one neural implicit SLAM, NICE-SLAM [58].

| Method | BA | Tracking | | Mapping |
|--------|-----|----------|----------|---------|
| | | ATE$_t$ (mm)↓ | ATE$_r$ (°)↓ | Chamfer (mm)↓ |
| NICE-SLAM, | Photo | 4.16 | 2.68 | 1.95 |
| *CVPR'22* | NFL | **2.88** | 2.81 | **1.70** |
| EndoGSLAM, | Photo | 1.93 | 1.81 | 0.85 |
| *MICCAI'24* | NFL | 2.04 | **1.13** | 0.97 |
| MonoGS, | Photo | 2.90 | 1.11 | 1.16 |
| *CVPR'24* | NFL | **1.60** | 1.49 | **0.79** |

## 5.1 Evaluation Setting

**Datasets.** We evaluate our method on three datasets that reflect different challenges in handling near-field dynamic lighting: (1) a phantom endoscopy dataset, (2) a clinical endoscopy dataset, and (3) a dataset of indoor scenes captured with phone camera with flashlight turned on.

*C3VD.* The C3VD dataset [4] (CC BY-NC-SA 4.0) was created using a phantom colon with synthetic materials to simulate realistic tissue geometry. *Colon10K.* To test generalization in real-world clinical endoscopy settings, we evaluate on Colon10K [24], a large-scale video dataset without depth or pose supervision. Videos are sampled from actual The endoscopy video was captured by a surgeon who performs different endoscopy procedures on the phantom colon with a real endoscope capturing RGB images coupled with corresponding depth maps. We focus on 8 sequences ranging from 70 to 800 frames from different regions of the colon, for more details, please see supplementary. We evalute using both ground truth and predicted depths.

Table 2: **Quantitative evaluation on the C3VD [4]** dataset using depth maps estimated by SOTA techniques, PPSNet [35] and DA-Hybrid [50, 32]. Replacing `Photometric BA` with NFL-BA significantly improves tracking for both MonoGS [27] and EndoGSLAM [45], and mapping and rendering quality for MonoGS [27]. Note that `SOTA` performance for each of the tracking, mapping, and rendering metrics is observed when NFL-BA is used.

| Method | Depth | BA | Tracking | | Mapping | Rendering |
| | | | $\text{ATE}_t$ (mm)↓ | $\text{ATE}_r^\circ$ ↓ | Chamfer (mm) ↓ | LPIPS ↓ |
| --- | --- | --- | --- | --- | --- | --- |
| EndoGSLAM [45] *MICCAI'24* | PPS-Net | Photo | 3.03 | 1.73 | 1.23 | 0.39 |
| | PPS-Net | NFL | **2.62** | **1.24** | 1.25 | 0.39 |
| | DA-Hybrid | Photo | 6.67 | 2.26 | 2.12 | 0.43 |
| | DA-Hybrid | NFL | **3.91** | **1.58** | 2.39 | **0.42** |
| MonoGS [27] *CVPR'24* | PPS-Net | Photo | 3.48 | 1.70 | 1.59 | 0.56 |
| | PPS-Net | NFL | **2.18** | **1.65** | **0.99** | **0.53** |
| | DA-Hybrid | Photo | 4.63 | 1.69 | 1.34 | 0.52 |
| | DA-Hybrid | NFL | **2.35** | **1.14** | **1.13** | 0.52 |

procedures and are typically around 300-600 frames. This setting is significantly more challenging than the phantom setting, since frames may contain motion blur, specular highlights, and fluid occlusions. Sequences are uniformly sampled and fisheye corrected.

*Self-Captured Indoor Scenes.* To study the role of near-field lighting in a controlled non-clinical setting, we capture a dataset of four indoor scenes (*Guitars*, *Porch*, *Pool*, and *Stairs*) using an iPhone 15 Pro. Scenes include objects with varied geometry and reflectance (diffuse, specular), imaged under dynamic motion. Scenes are captured using a co-located point light source mounted to camera. Ground-truth camera trajectories were recorded via motion capture, but no reference point clouds are available; hence, we report only trajectory error ($\text{ATE}_t$) and perceptual quality (LPIPS), omitting Chamfer distance. Details and additional visualizations are included in the supplementary.

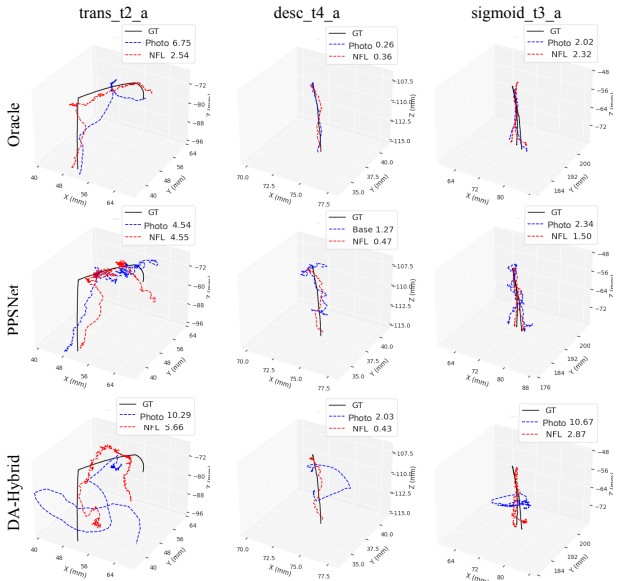

Figure 5: **Camera tracking improvement over En-doGSLAM [45].** Replacing the Photo-BA loss (in blue) with NFL-BA loss (in red) significantly improves camera tracking for different depth initialization. Average tracking error $\text{ATE}_t$ for each sequence is reported in the inset. (zoom for details)

**Metrics.** For evaluation, we basically followed other neural rendering SLAM algorithms [27, 45]. For tracking performance, we measure the root mean square error of the Absolute Trajectory Error (ATE) for both translation and rotation across all frames. Translation error $\text{ATE}_t$ is in millimeters (mm) for the endoscopy scenes and meters (m) for the in-door scenes. And rotation error $\text{ATE}_r$ is in degrees. To assess the mapping quality, we use the Chamfer distance from ground truth point clouds to the nearest points in the estimated point clouds [46], for more details please see supplementary. In addition, we evaluate rendering quality using the Learned Perceptual Image Patch Similarity (LPIPS) [54]. We note that for many endoscopic SLAM applications, tracking and mapping accuracies are more important than photorealism of the rendered images, unlike many indoor or outdoor scenes.

**Computational costs** We trained all models on a single NVIDIA RTX A6000 GPU. The per-scene optimization takes ~1 FPS. For more information on runtime speed, please see supplementary materials.

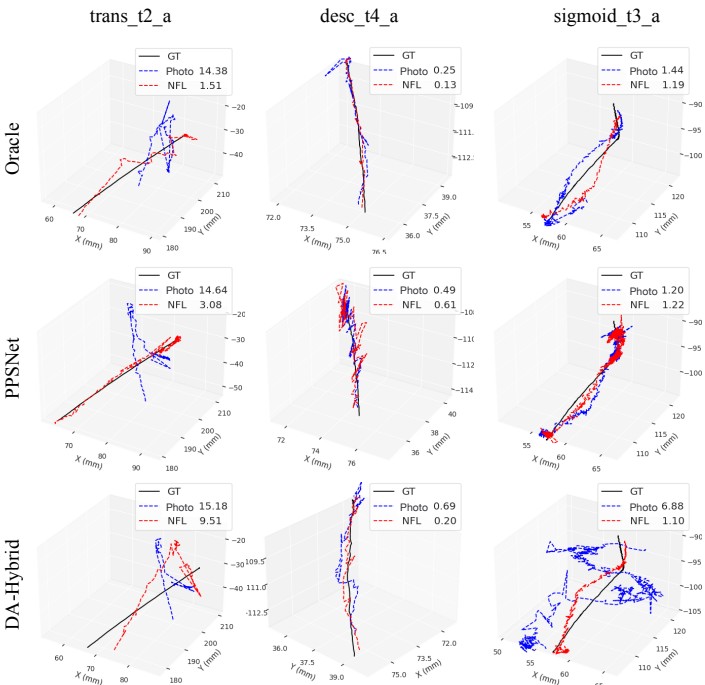

Figure 6: **Camera tracking improvement over MonoGS [27].** Replacing the Photo BA loss (in blue) with NFL-BA loss (in red) significantly improves camera tracking for different depth initialization. Average tracking error $\text{ATE}_t$ for each sequence is reported in the inset.

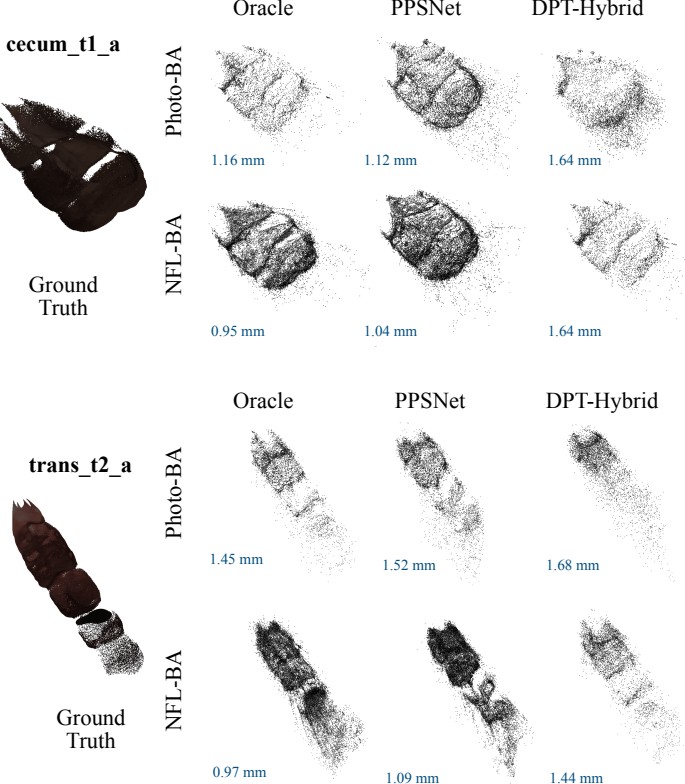

Figure 7: **Reconstructed point clouds using MonoGS [27]** show that NFL-BA improves coverage and density while reducing scatter compared to Photometric BA, as measured by Chamfer distance.

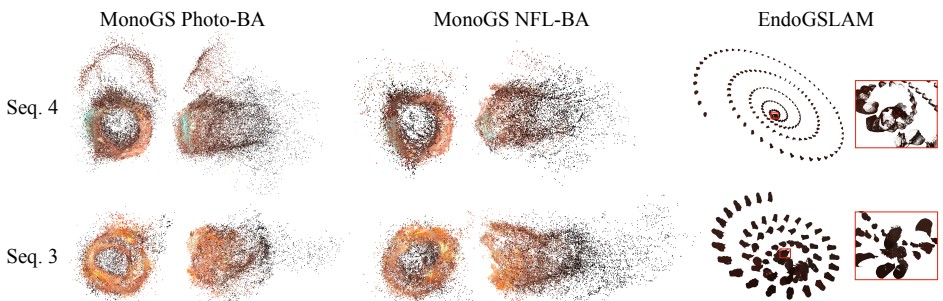

MonoGS Photo-BA      MonoGS NFL-BA      EndoGSLAM

Seq. 4

Seq. 3

Figure 8: **Results on real endoscopy from Colon10k dataset.** On Sequences 3 and 4 with PPSNet depth, NFL-BA improves MonoGS tracking and mapping, yielding more coherent, elongated colon structures, while EndoGSLAM fails under sudden camera motion.

## 5.2 Evaluation on C3VD Endoscopy Data

Because NFL-BA is designed as a drop-in replacement for photometric bundle adjustment, we only adjusted the two associated loss weights; all other hyperparameters remain identical between the Photo-BA and NFL-BA experiments. Please see supplemental for detailed hyperparameter settings.

**SLAM with oracle depth map.** In Tab. 1 we replace Photometric Bundle Adjustment loss with NFL-BA loss for depth is initialized with ground-truth or oracle. NFL-BA significantly improves camera localization ($ATE_t$) and mapping for NICE-SLAM and MonoGS, and only camera rotation ($ATE_r$) for EndoGSLAM. EndoGSLAM was specifically designed for synthetic data with an oracle depth map, and we will show later that for estimated depth maps or real endoscopy videos, it performs significantly worse than MonoGS Ground-truth depths are never available during endoscopy, and the majority of endoscopes hardly have any depth sensors.

**SLAM with predicted depth map.** Under realistic conditions with estimated depths, NFL-BA's impact is even more pronounced. In Tab. 2 we replace Photometric BA loss with NFL-BA loss for MonoGS [27] and EndoGSLAM [45] for depth maps we use PPSNet [35], a state-of-the-art monocular depth estimation algorithm for endoscopy, and fine-turned general-purpose depth estimator, which we will call it as DA-Hybrid - DepthAnything[50] with DINOv2 encoder [32]. NFL-BA significantly improves camera localization ($ATE_t$) and camera rotation ($ATE_r$) for both MonoGS [27] and EndoGSLAM [45] while producing similar rendering quality. For example, camera localization for MonoGS is improved by 37% for PPSNet and 49% for DA-Hybrid depth initialization. Mapping accuracy of MonoGS also improves by 37% for PPSNet and 16% for DA-Hybrid depth maps. Overall, these results demonstrate that NFL-BA can compensate for noisy depth estimation and improves performance. Across all four metrics, for tracking, mapping, and rendering, the SOTA performance on the C3VD dataset is in fact achieved when NFL-BA loss is used in the SLAM framework.

## 5.3 Evaluation on Real Endoscopy Data

We show results on real endoscopy sequence from Colon10k sequence 3 and 4 in Fig. 8. EndoGSLAM fails to construct any real structure, with many disconnected regions along a spiral trajectory. EndoGSLAM assumes constant velocity and is not robust to the sudden motion common in endoscopy procedures, which is significantly more in real data than C3VD. This results extremely poor or failed reconstructions.

**Sequence 4.** This pull-back "down-the-barrel" sequence exposes a clear cylindrical lumen. With Photo-BA, MonoGS captures the overall shape but produces a broken segment due to trajectory drift. NFL-BA corrects this, yielding a continuous "hollow-center" reconstruction. Minor artifacts from extreme specular highlights remain (green points), as detailed in the supplement.

**Sequence 3.** In the extended traversal, both Photo-BA and NFL-BA recover the colon's general geometry, but NFL-BA produces a longer, tighter model with less point scatter. It also better preserves interior ridges (interactive point clouds in the supplement).

## 5.4 Evaluation on Indoor Scene

To validate NFL-BA in a non-medical setting, we evaluate on four indoor scenes. Table 3 shows that replacing standard Photometric BA with NFL-BA yields substantial reductions in $ATE_t$ across

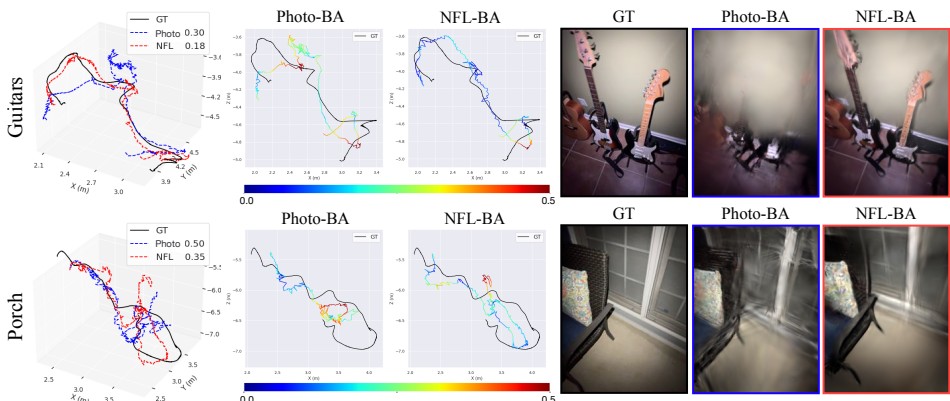

Figure 9: **Results on indoor scenes captured with co-located flashlight and phone camera.** Qualitative comparison on two self-captured indoor scenes using MonoGS with standard Photo-BA versus NFL-BA. (left) Estimated camera trajectories overlaid on ground truth. (center) Per-frame tracking error relative to ground truth. (right) Example re-rendered views, illustrating the sharper, more accurate reconstructions enabled by NFL-BA.

Table 3: Quantitative results on four self-captured indoor scenes under dynamic lighting, comparing MonoGS with standard Photo-BA versus NFL-BA. For each scene, the best of each metric is bold.

| BA | Guitars | | Porch | | Pool | | Stairs | |
|---|---|---|---|---|---|---|---|---|
| | $ATE_t$ (m)↓ | LPIPS ↓ | $ATE_t$ (m)↓ | LPIPS ↓ | $ATE_t$ (m)↓ | LPIPS ↓ | $ATE_t$ (m)↓ | LPIPS ↓ |
| Photo | 0.30 | 0.39 | 0.50 | **0.49** | 0.41 | 0.46 | 0.36 | 0.40 |
| NFL | **0.18** | **0.37** | **0.35** | 0.50 | **0.30** | **0.44** | **0.20** | **0.31** |

all scenes: from 0.30m to 0.18m (40%) in *Guitar*, 0.50m to 0.35m (30%) in *Outdoor*, 0.41m to 0.30m (27%) in *Pool*, and 0.36m to 0.20m (44%) in *Stair*. On average, NFL-BA reduces tracking error by $\sim 35\%$, demonstrating that near-field shading cues greatly enhance pose estimation even in richly textured, well-lit indoor environments. While LPIPS remains largely comparable, with slight improvements in *Guitars* and *Stairs* and minor variations in *Porch* and *Pool*, the primary benefit of NFL-BA is clear in trajectory accuracy (see Fig. 9).

## 6 Conclusions

In this paper, we presented a novel bundle adjustment loss that explicitly models dynamic near-field lighting by incorporating light intensity fall-off based on the relative distance and orientation between the surface and the co-located light and camera. This formulation is especially effective for endoscopic scenes, where traditional geometric or photometric bundle adjustment losses struggle under dynamic near-field lighting conditions on textureless surfaces. We demonstrated the general applicability of our approach by integrating it into three different neural rendering-based SLAM methods, improving performance on a challenging endoscopy dataset and indoor scenes captured with a phone camera with a flashlight turned on.

**Limitations.** While our new formulation for SLAM effectively represents scenes with co-located and dynamic lighting environments, it is currently limited in handling specular reflections, sub-surface scattering, and inter-reflections. Incorporating a more complex image formulation is beyond the scope of the current work, and addressing these remains a promising direction for future research.

## 7 Acknowledgments

This work is supported by a National Institute of Health (NIH) project #R21EB035832 "Next-gen 3D Modeling of Endoscopy Videos" and #R21EB037440 "Gen-AI Airway Simulator for 3D Endoscopy". We also thank Stephen M. Pizer, Ron Alterovitz, and Dr. Sarah McGill for helpful discussions during the project.

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

# Supplementary Materials

## Overview of Appendices

Our appendices contain the following additional details:

- Appendix A: Dataset Processing and Collection
- Appendix B: Implementation Details and Computational Costs
- Appendix C: Point Clouds and Metrics
- Appendix D: Long Sequence Validation
- Appendix E: Evaluation of Center-Crop Baseline
- Appendix F: Lighting model and Angular Attenuation
- Appendix G: Results on Endoscopy Datasets

## A  Dataset Processing and Collection

**C3VD.** We test our method on a subset of 8 videos with at least one video from each section of the colon, with varying camera motion, and anomalies. We choose these 8 videos from the test split of PPSNet to avoid any bias when the SLAM is initialized with PPSNet predicted depth map. The names of the sequences are as follows: cecum_t1_a, cecum_t2_a, cecum_t3_a, sigmoid_t3_a, desc_t4_a_p2, trans_t2_a, trans_t3_a, and trans_t4_a.

All images were cropped to remove any artifacts resulting from fish-eye correction then downscale and crop the images. Specifically, we resize each image to have a height of 384 pixels while maintaining the aspect ratio, then crop the central region to obtain a 384×384 pixel image.

**Colon10k.** Since Colon10K provides no ground-truth depths, we compute per-frame estimates with PPSNet. Each image is center-cropped and uniformly resized to $384 \times 384$ px to match our SLAM input requirements.

**Indoor Self Captures.** We recorded four indoor scenes using an iPhone 15 with LiDAR, capturing synchronized RGB ($1440 \times 1920$ px) and depth ($256 \times 192$ px) streams. Raw RGB frames are downscaled to $256 \times 192$ px to align with the depth map resolution. Depth maps are stored as 16-bit values up to 10 m. We logged camera poses via Apple's ARKit framework, code for our custom capture app and preprocessing scripts will be released alongside the dataset.

## B  Implementation Details and Computational Costs

As mentioned in the main paper, we ran all models on a single NVIDIA RTX A6000 GPU. The per-scene optimization takes approximately 1 FPS. We found that NFL-BA only reduces the fps runtime by a small amount on the C3VD dataset.

**NICE-SLAM.** Since the scene is encoded using neural networks, we extract normals from the occupancy grid, as described in Sec. 4.2 to calculate the shading term. NICE-SLAM requires a well-defined bounding box which we obtained from the ground truth point clouds (see appendix C). We also used the default loss weights of NICE-SLAM, setting $\lambda_{ren}$ to 0.5 during tracking and 0.2 during mapping, and $\lambda_{geo}$ to 1 in both phases.

**EndoGSLAM.** The main difference is the weight map $M_t$ in the bundle adjustment loss (Eq. 3) to exclude over-exposed pixels that can arise in endoscopy-specific lighting conditions. Furthermore, given that the shading term $PPS(\cdot)$ is sensitive to depth scales, we rescaled the depth maps so that their maximum values are approximately 5. Notably, scaling the depth maps did not improve baseline performance (when using PPS depth maps, the average $ATE_t$ went from 3.03 to 3.38). We used the

Table 4: Runtime: We evaluate the frames per second (**FPS**) for all methods on the C3VD dataset.

| Method | Depth | Photo-BA | NFL-BA |
|--------|-------|----------|--------|
| NICE-SLAM | Oracle | $\ll 1$ | $\ll 1$ |
| | PPSNet | $\ll 1$ | $\ll 1$ |
| EndoGSLAM | Oracle | 1.79 | 1.35 |
| | PPSNet | 1.53 | 1.22 |
| | DPT-Hybrid | 1.20 | 0.90 |
| MonoGS | Oracle | 1.38 | 1.09 |
| | PPSNet | 1.06 | 0.93 |
| | DPT-Hybrid | 0.99 | 0.83 |

default loss weights of EndoGSLAM, $\lambda_{ren}$; $\lambda_{geo}$, set to 0.5 and 1 during tracking and 1 and 1 during mapping, respectively.

**MonoGS.** To integrate our method, we treat the Gaussian color features as albedo features and multiply them with the shading term before rasterization, and then we use the rendered output colors for bundle adjustment. For all input depths, we set $\lambda_{ren}$ and $\lambda_{geo}$ to 0.8 and 0.5, respectively.

## C  Point Clouds and Metrics

**Chamfer Distances.** Since ground truth point clouds are unavailable for C3VD, we generate them by unprojecting 2D images into 3D space with the correct camera configuration and the oracle depth maps, provided in the C3VD dataset. For neural fields-based SLAM, we use the vertices of the output meshes as the estimated point clouds, while for Gaussian Splatting-based SLAMs, we use the Gaussian positions. For point cloud alignment, we use Coherent Point Drift [31] and the Chamfer distances are also in millimeters.

**Coloring Point Clouds.** We use extracted point clouds from 3D Gaussian positions for visualization. For colors, we directly use Gaussian color features.

## D  Long Sequence Validation

To evaluate NFL-BA's performance on longer sequence, we test our method on a longer screening video for C3VD, specifically **c0_full_t2_v2**, which spans over 4,000 frames with ground-truth poses, allowing us to measure cumulative drift. In the main paper, we only evaluate on registered videos with ground truth depth, not screening videos. We show how NFL-BA performs relative to Photo-BA using the MonoGS backbone in Table 5.

NFL-BA matches Photometric BA on short sequences and increasingly outperforms it as trajectory length grows. As a plug-and-play bundle-adjustment loss, NFL-BA enhances long-term robustness under dynamic near-field lighting.

Table 5: ATE_T (cm) on c0_full_t2_v2 at varying lengths. NFL-BA matches or beats Photo-BA on short sequences and shows long-term stability.

| BA | 500 frames | 1,000 frames | 2,000 frames | 4,000 frames |
|----|-----------|--------------|--------------|--------------|
| Photo | 2.599 | 1.937 | 8.459 | 14.790 |
| NFL | 2.422 | 2.254 | 5.742 | 11.368 |

## E  Evaluation of Center-Crop Baseline

We compared MonoGS Photo-BA optimized on only the central 75% and 50% of each frame against full-frame MonoGS + NFL-BA on two sequences, measuring translational ATE ($ATE_T$), Chamfer distance (CD), and reconstructed point count in Table 6.

Although center-cropping improves camera tracking performance (ATE_T) of MonoGS and gets close to NFL-BA, it results in significantly worse reconstruction in terms of quality (CD) and density (number of points). This is because for camera localization, not all pixels are essential, and focusing only on central pixels can eliminate near-field lighting effects. In contrast, for reconstruction, all pixels matter. This highlights the need for a principled mechanism for handling dynamic near-field lighting, as proposed by NFL-BA.

Table 6: Center-cropping improves trajectory error but reduces map density, while full-frame NFL-BA maintains both accuracy and dense reconstructions.

| Method | trans_t3_a | | | desc_t4_p2 | | |
|---|---|---|---|---|---|---|
| | $ATE_T$ | CD | Points | $ATE_T$ | CD | Points |
| MonoGS + NFL-BA | 0.26 | 0.60 | 28,196 | 0.13 | 0.67 | 8,879 |
| MonoGS (full frame) | 0.31 | 0.76 | 7,095 | 0.25 | 0.76 | 6,398 |
| MonoGS (75% center) | 0.35 | 1.06 | 5,761 | 0.13 | 0.97 | 4,329 |
| MonoGS (50% center) | 0.40 | 2.82 | 1,865 | 0.15 | 1.56 | 2,100 |

## F  Lighting Model and Angular Attenuation

We clarify that NFL-BA models only diffuse reflectance and direct illumination, ignoring specular and subsurface effects. Explicitly modeling these introduces non-differentiable and highly non-convex terms, remaining an open challenge.

To mitigate specular highlights, we apply an intensity mask discarding pixels above 0.9 grayscale intensity. This handles most artifacts but cannot correct reflective surfaces beyond the mask. While effective on matte datasets (C3VD, in-the-wild), degradation occurs on Colon10K with more specular highlights. Future work will address specular reflections under dynamic lighting.

Regarding angular attenuation, $\beta = 0$ is justified for tightly collimated endoscopic LEDs, we must validated this for non endoscopy scenes. Using NFL-BA on the MonoGS backbone, we validate various $\beta$ values on two indoor sequences as seen in Table 7

$\beta = 0$ provides competitive mean accuracy and low variance. Non-zero values may yield small gains but introduce instability. Given minimal benefit versus tuning cost, we retain $\beta = 0$, and plan to explore learning $\beta$ as a per-scene parameter in future work.

Table 7: Ablation of the angular-attenuation coefficient $\beta$ on two indoor sequences, reporting translational ATE and LPIPS.

| Scene | Metric | $\beta$=0.00 | $\beta$=0.25 | $\beta$=0.50 | $\beta$=0.75 | $\beta$=1.00 |
|---|---|---|---|---|---|---|
| Guitars | ATE_T | 0.17±0.01 | 0.17±0.003 | 0.16±0.006 | 0.17±0.006 | 0.17±0.011 |
| | LPIPS | 0.36±0.02 | 0.37±0.006 | 0.40±0.030 | 0.36±0.017 | 0.36±0.020 |
| Porch | ATE_T | 0.33±0.16 | 0.25±0.013 | 0.27±0.045 | 0.36±0.158 | 0.22±0.012 |
| | LPIPS | 0.50±0.03 | 0.53±0.008 | 0.48±0.005 | 0.51±0.035 | 0.49±0.010 |

## G  Results on Endoscopy Datasets

For all experiments for each of the slam systems, we report the median of three runs for all tables and figures. We have included per-sequence metrics for the median run below. Additionally, we include point clouds for EndoGSLAM in Figure 10, more can be found on our project page.

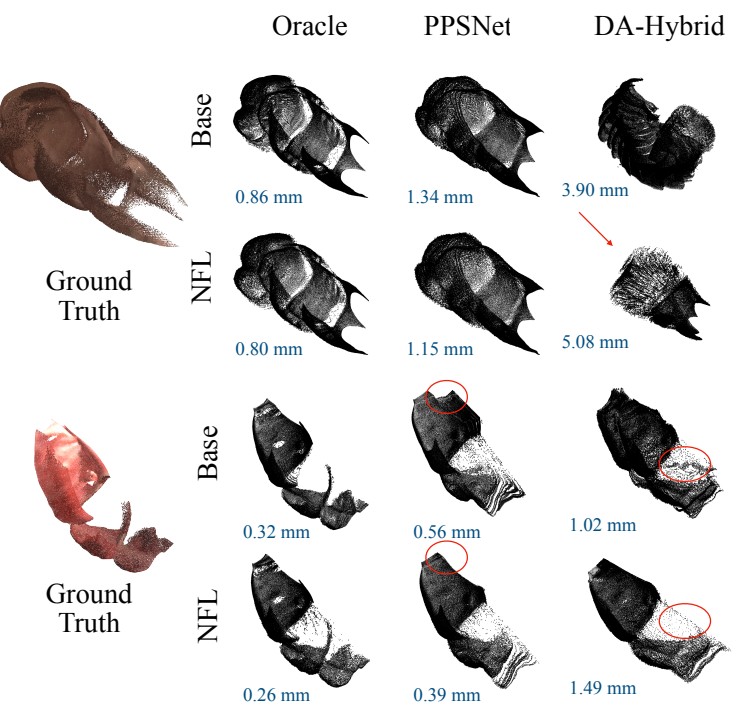

Figure 10: EndoGSLAM Point cloud results for 2 sequences: cecum_t1_a_under_review (top) and desc_t4_a_p2_under_review (bottom).

Table 8: Results for sequence **cecum_t1_a_under_review**

| Method | Depth | BA | ATE$_t$ (mm)↓ | ATE$_r$(°)↓ | Chamfer (mm)↓ |
|---|---|---|---|---|---|
| NICE-SLAM | Oracle | Photo | 2.65 | 2.82 | 0.08 |
| | | NFL | 1.39 | 2.81 | 0.15 |
| | PPS-Net | Photo | 10.14 | 2.71 | 0.51 |
| | | NFL | 4.14 | 2.75 | 0.19 |
| EndoGSLAM | Oracle | Photo | 1.39 | 0.24 | 0.12 |
| | | NFL | 0.91 | 0.31 | 0.16 |
| | PPS-Net | Photo | 2.79 | 0.60 | 0.30 |
| | | NFL | 2.93 | 0.70 | 0.35 |
| | DA-Hybrid | Photo | 2.64 | 0.08 | 0.04 |
| | | NFL | 8.44 | 2.42 | 1.21 |
| MonoGS | Oracle | Photo | 1.16 | 0.39 | 1.56 |
| | | NFL | 1.22 | 0.35 | 0.95 |
| | PPS-Net | Photo | 2.30 | 0.65 | 1.19 |
| | | NFL | 2.45 | 0.76 | 1.04 |
| | DA-Hybrid | Photo | 5.44 | 0.30 | 1.64 |
| | | NFL | 1.09 | 0.40 | 1.65 |

Table 9: Results for sequence **cecum_t2_a_under_review**

| Method | Depth | BA | ATE$_t$ (mm)↓ | ATE$_r$(°)↓ | Chamfer (mm)↓ |
|---|---|---|---|---|---|
| NICE-SLAM | Oracle | Photo | 8.13 | 2.19 | 0.49 |
| | | NFL | 1.15 | 2.82 | 0.11 |
| | PPS-Net | Photo | 1.11 | 2.13 | 0.11 |
| | | NFL | 7.98 | 2.82 | 0.50 |
| EndoGSLAM | Oracle | Photo | 2.32 | 1.06 | 0.53 |
| | | NFL | 6.75 | 2.79 | 1.40 |
| | PPS-Net | Photo | 4.55 | 1.18 | 0.59 |
| | | NFL | 4.55 | 2.82 | 1.41 |
| | DA-Hybrid | Photo | 5.66 | 0.89 | 0.45 |
| | | NFL | 9.47 | 2.25 | 1.13 |
| MonoGS | Oracle | Photo | 4.04 | 0.34 | 1.47 |
| | | NFL | 6.84 | 0.48 | 1.24 |
| | PPS-Net | Photo | 6.72 | 2.71 | 1.74 |
| | | NFL | 6.52 | 2.73 | 1.70 |
| | DA-Hybrid | Photo | 5.20 | 0.66 | 2.18 |
| | | NFL | 4.26 | 0.32 | 1.35 |

Table 10: Results for sequence **cecum_t3_a_under_review**

| Method | Depth | BA | ATE$_t$ (mm)↓ | ATE$_r$(°)↓ | Chamfer (mm)↓ |
|---|---|---|---|---|---|
| NICE-SLAM | Oracle | Photo | 3.46 | 2.82 | 0.11 |
| | | NFL | 3.42 | 2.82 | 0.07 |
| | PPS-Net | Photo | 2.80 | 2.64 | 0.17 |
| | | NFL | 3.83 | 2.59 | 0.14 |
| EndoGSLAM | Oracle | Photo | 0.75 | 0.15 | 0.08 |
| | | NFL | 0.74 | 0.27 | 0.14 |
| | PPS-Net | Photo | 0.79 | 0.16 | 0.08 |
| | | NFL | 2.18 | 0.22 | 0.11 |
| | DA-Hybrid | Photo | 1.45 | 0.62 | 0.31 |
| | | NFL | 1.52 | 0.18 | 0.09 |
| MonoGS | Oracle | Photo | 0.36 | 0.16 | 1.23 |
| | | NFL | 0.87 | 0.31 | 0.62 |
| | PPS-Net | Photo | 1.07 | 0.17 | 1.12 |
| | | NFL | 1.24 | 0.14 | 0.81 |
| | DA-Hybrid | Photo | 1.06 | 0.33 | 0.95 |
| | | NFL | 1.16 | 0.30 | 0.93 |

Table 11: Results for sequence **desc_t4_a_p2_under_review**

| Method | Depth | BA | ATE$_t$ (mm)↓ | ATE$_r$(°)↓ | Chamfer (mm)↓ |
|--------|-------|-----|------|------|------|
| NICE-SLAM | Oracle | Photo | 15.90 | 2.65 | 0.59 |
| | | NFL | 8.88 | 2.75 | 0.50 |
| | PPS-Net | Photo | 0.87 | 2.80 | 0.10 |
| | | NFL | 0.68 | 2.80 | 0.10 |
| EndoGSLAM | Oracle | Photo | 0.36 | 0.56 | 0.28 |
| | | NFL | 0.26 | 0.94 | 0.47 |
| | PPS-Net | Photo | 0.48 | 0.58 | 0.29 |
| | | NFL | 1.00 | 0.83 | 0.42 |
| | DA-Hybrid | Photo | 0.45 | 1.16 | 0.58 |
| | | NFL | 1.80 | 2.81 | 1.40 |
| MonoGS | Oracle | Photo | 0.25 | 1.05 | 0.76 |
| | | NFL | 0.13 | 0.98 | 0.67 |
| | PPS-Net | Photo | 0.49 | 1.78 | 0.77 |
| | | NFL | 0.61 | 1.67 | 0.70 |
| | DA-Hybrid | Photo | 0.69 | 2.63 | 0.80 |
| | | NFL | 0.20 | 0.97 | 0.77 |

Table 12: Results for sequence **sigmoid_t3_a_under_review**

| Method | Depth | BA | ATE$_t$ (mm)↓ | ATE$_r$(°)↓ | Chamfer (mm)↓ |
|--------|-------|-----|------|------|------|
| NICE-SLAM | Oracle | Photo | 1.04 | 2.47 | 0.05 |
| | | NFL | 0.84 | 2.83 | 0.05 |
| | PPS-Net | Photo | 9.47 | 2.79 | 0.28 |
| | | NFL | 6.60 | 2.55 | 0.35 |
| EndoGSLAM | Oracle | Photo | 4.81 | 1.14 | 0.57 |
| | | NFL | 4.11 | 2.69 | 1.35 |
| | PPS-Net | Photo | 3.13 | 1.40 | 0.70 |
| | | NFL | 9.82 | 2.56 | 1.29 |
| | DA-Hybrid | Photo | 8.51 | 2.36 | 1.18 |
| | | NFL | 8.15 | 2.80 | 1.41 |
| MonoGS | Oracle | Photo | 1.44 | 0.46 | 1.17 |
| | | NFL | 1.19 | 2.33 | 0.67 |
| | PPS-Net | Photo | 1.20 | 1.53 | 4.63 |
| | | NFL | 1.22 | 2.22 | 0.89 |
| | DA-Hybrid | Photo | 6.88 | 2.18 | 1.42 |
| | | NFL | 1.10 | 0.67 | 1.33 |

Table 13: Results for sequence **trans_t2_a_under_review**

| Method | Depth | BA | $\text{ATE}_t$ (mm)↓ | $\text{ATE}_r$(°)↓ | Chamfer (mm)↓ |
|---|---|---|---|---|---|
| NICE-SLAM | Oracle | Photo | 0.77 | 2.83 | 0.05 |
| | | NFL | 0.94 | 2.80 | 0.05 |
| | PPS-Net | Photo | 9.18 | 2.82 | 0.26 |
| | | NFL | 9.85 | 2.81 | 0.31 |
| EndoGSLAM | Oracle | Photo | 4.21 | 1.80 | 0.90 |
| | | NFL | 0.49 | 2.82 | 1.41 |
| | PPS-Net | Photo | 10.05 | 1.66 | 0.84 |
| | | NFL | 0.90 | 1.46 | 0.73 |
| | DA-Hybrid | Photo | 9.14 | 2.77 | 1.39 |
| | | NFL | 14.59 | 2.18 | 1.10 |
| MonoGS | Oracle | Photo | 14.38 | 1.42 | 1.49 |
| | | NFL | 1.51 | 2.72 | 0.97 |
| | PPS-Net | Photo | 14.64 | 1.91 | 1.52 |
| | | NFL | 3.08 | 1.02 | 1.09 |
| | DA-Hybrid | Photo | 15.18 | 2.12 | 1.68 |
| | | NFL | 9.51 | 1.41 | 1.45 |

Table 14: Results for sequence **trans_t3_a_under_review**

| Method | Depth | BA | $\text{ATE}_t$ (mm)↓ | $\text{ATE}_r$(°)↓ | Chamfer (mm)↓ |
|---|---|---|---|---|---|
| NICE-SLAM | Oracle | Photo | 0.97 | 2.81 | 0.15 |
| | | NFL | 6.26 | 2.79 | 0.68 |
| | PPS-Net | Photo | 3.78 | 2.80 | 0.22 |
| | | NFL | 1.12 | 2.65 | 0.13 |
| EndoGSLAM | Oracle | Photo | 0.17 | 2.70 | 1.35 |
| | | NFL | 0.20 | 2.70 | 1.35 |
| | PPS-Net | Photo | 0.30 | 2.80 | 1.40 |
| | | NFL | 0.50 | 2.81 | 1.41 |
| | DA-Hybrid | Photo | 0.46 | 2.81 | 1.41 |
| | | NFL | 0.33 | 2.74 | 1.37 |
| MonoGS | Oracle | Photo | 0.31 | 2.67 | 0.76 |
| | | NFL | 0.26 | 2.66 | 0.60 |
| | PPS-Net | Photo | 0.33 | 2.79 | 0.89 |
| | | NFL | 0.61 | 2.83 | 0.82 |
| | DA-Hybrid | Photo | 0.29 | 2.81 | 0.91 |
| | | NFL | 0.38 | 2.71 | 0.71 |

Table 15: Results for sequence **trans_t4_a_under_review**

| Method | Depth | BA | ATE$_t$ (mm)↓ | ATE$_r$(°)↓ | Chamfer (mm)↓ |
|---|---|---|---|---|---|
| NICE-SLAM | Oracle | Photo | 0.39 | 2.81 | 0.05 |
| | | NFL | 0.23 | 2.83 | 0.05 |
| | PPS-Net | Photo | 7.26 | 2.77 | 0.14 |
| | | NFL | 7.88 | 2.73 | 0.23 |
| EndoGSLAM | Oracle | Photo | 2.64 | 1.47 | 0.74 |
| | | NFL | 1.99 | 2.02 | 1.01 |
| | PPS-Net | Photo | 2.02 | 1.73 | 0.86 |
| | | NFL | 2.58 | 1.78 | 0.89 |
| | DA-Hybrid | Photo | 2.99 | 2.08 | 1.04 |
| | | NFL | 10.67 | 2.72 | 1.37 |
| MonoGS | Oracle | Photo | 1.23 | 2.41 | 0.83 |
| | | NFL | 0.76 | 2.06 | 0.62 |
| | PPS-Net | Photo | 1.12 | 2.10 | 0.89 |
| | | NFL | 1.70 | 1.80 | 0.86 |
| | DA-Hybrid | Photo | 2.30 | 2.46 | 1.10 |
| | | NFL | 1.14 | 2.36 | 0.85 |

