# OpenReview forum: "NFL-BA: Near-Field Light Bundle Adjustment for SLAM in Dynamic Lighting"
_NeurIPS.cc/2025/Conference — NeurIPS 2025 poster_

### Official Review · Reviewer_BMQ8 · 2025-06-27

**Clarity:** 3
**Significance:** 3
**Originality:** 3
**Rating:** 5
**Confidence:** 5

**Summary:**

This paper proposes a novel Near-Field Lighting Bundle Adjustment (NFL-BA) loss to address the performance degradation of SLAM systems under dynamic near-field lighting conditions (e.g., endoscopy, underground exploration). By explicitly integrating a near-field lighting attenuation model into the bundle adjustment loss of neural rendering-based SLAM, our method decouples surface albedo from lighting intensity, significantly improving the accuracy of camera pose estimation and scene reconstruction. On the C3VD colonoscopy dataset, NFL-BA reduces MonoGS’s tracking error by 37% (Table 2) and achieves an average 35% improvement in pose estimation in indoor scenes captured with smartphone flashlights (Table 3). Experiments demonstrate that NFL-BA is a universal and effective plug-and-play solution.

**Questions:**

1. How does NFL-BA address specular reflections (e.g., from intestinal fluid/mucus) that violate the pure diffuse assumption?
2. Why is ignoring angular attenuation (β=0) valid for smartphone flashlights? Does this overestimate irradiance near surfaces?
3, How robust is NFL-BA without depth inputs (e.g., monocular RGB-only) or in low-texture/low-light scenes?

**Ethical Concerns:**

["NO or VERY MINOR ethics concerns only"]

**Final Justification:**

One core issue I concerned have been addressed.

**Limitations:**

As mentioned by authors, it is currently limited in handling specular reflections, sub-surface scattering, and inter-reflections.

**Quality:**

3

**Strengths And Weaknesses:**

Strengths:
1. This work systematically tackles dynamic lighting interference in SLAM scenarios where the camera and light source move together (e.g., endoscopy, rescue robots), addressing a critical gap in traditional SLAM methods that assume static far-field lighting. The proposed lighting-geometric decoupling model separates surface albedo from near-field lighting attenuation (inverse-square law and normal vector angle), enhancing robustness to dynamic shadows (Equations 5–7).

2. NFL-BA seamlessly integrates into implicit (e.g., NICE-SLAM) and explicit (e.g., 3D Gaussian Splatting) neural rendering SLAM frameworks by replacing the Photometric BA loss (Equation 3) without requiring modifications to other components. Its versatility and practicality are validated across medical datasets (C3VD, Colon10K) and general scenarios (smartphone flashlight indoor scenes).

3. The experiments are comprehensive, with clear visualizations. On the C3VD dataset, NFL-BA consistently outperforms baselines: MonoGS’s tracking error drops by 37% (3.48 mm → 2.18 mm), and EndoGSLAM’s rotational error decreases by 38% (1.81° → 1.13°) (Tables 1–2). Real-world tests on colonoscopy data (Colon10K) and dynamic indoor lighting scenarios (Figure 7, Table 3) further confirm its practicality, with an average 35% reduction in pose errors.

Weaknesses:
1. The current dynamic lighting model has limitations. For instance, the model described in Section 4.1 (Equations 5–7) is based on a pure diffuse reflection assumption. However, specular highlights caused by intestinal fluid and mucus (e.g., reflections on mucosal surfaces) are not accounted for. This may lead to brightness distortion in liquid-containing regions, where specular highlights could be misinterpreted as high albedo.

2. While setting β=0 is reasonable for endoscopic scenarios (where inverse-square attenuation dominates), it remains unclear whether ignoring angular attenuation in wide-angle light sources (e.g., smartphone flashlights) could overestimate near-surface irradiance and compromise pose-tracking accuracy. Further clarification is needed.

3. Experimentally, the robustness of NFL-BA in extreme scenarios requires additional validation. For example, Table 2 could include pure monocular SLAM experiments without depth input (e.g., using RGB only). Additionally, Tables 1–3 should report error ranges from multiple trials (e.g., “2.18±0.15 mm”) to demonstrate statistical stability.

---

> ### Author Rebuttal · Authors · 2025-07-31
>
> ## Lighting Model Limitations
>
> We agree with the reviewer, and we clarified in the paper that the NFL-BA formulation only models diffuse reflectance and direct illumination, ignoring specular reflectance, subsurface scattering and inter-reflections. While these effects are important, physically modeling them in an optimization framework is **significantly challenging and an open problem** in the community. This is because **these effects introduce non-differentiable elements** (e.g., path-tracing for inter-reflections), **highly non-convex functions** (e.g., sub-surface scattering and specular reflections modelled as Spatially Varying BRDF), and additional parameters for material modeling.
>
> During experimentation on real data with specular reflections (both in endoscopy and in-the-wild captures), we first **detect and exclude specular regions** via a simple intensity mask. We convert each frame to grayscale and discard pixels above a 0.9 intensity threshold, filtering out bright highlights. While this approach handles most specular artifacts, it cannot correct for reflective surfaces beyond the mask. The C3VD phantom dataset and self-captured in-the-wild dataset have predominantly matte surfaces, so this works well. However, as noted, the Colon10K clinical endoscopy dataset contains significantly more specular highlights, leading to moderate degradation in reconstruction. We believe future research is required to explicitly account for specular reflections in the presence of dynamic lighting.
>
> ## Angular Attenuation
>
> We appreciate the reviewer’s point that setting ignoring angular attenuation (β=0) may overestimate irradiance under wide-angle lighting such as smartphone flashlights. While β=0 is physically justified for tightly collimated endoscopic LEDs, we validated this choice on two self-captured indoor sequences with varying β.
>
> Table A: Ablation of the angular-attenuation coefficient on two self-captured indoor sequences, reporting translational ATE and LPIPS across β ∈ {0.00, 0.25, 0.50, 0.75, 1.00}.
> | Scene   | Metric |    β = 0.00 |     β = 0.25 |     β = 0.50 |     β = 0.75 |     β = 1.00 |
> | ------- | ------ | ----------: | -----------: | -----------: | -----------: | -----------: |
> | Guitars | ATE\_T | 0.17 ± 0.01 | 0.17 ± 0.003 | 0.16 ± 0.006 | 0.17 ± 0.006 | 0.17 ± 0.011 |
> |         | LPIPS  | 0.36 ± 0.02 | 0.37 ± 0.006 | 0.40 ± 0.030 | 0.36 ± 0.017 | 0.36 ± 0.020 |
> | Porch   | ATE\_T | 0.33 ± 0.16 | 0.25 ± 0.013 | 0.27 ± 0.045 | 0.36 ± 0.158 | 0.22 ± 0.012 |
> |         | LPIPS  | 0.50 ± 0.03 | 0.53 ± 0.008 | 0.48 ± 0.005 | 0.51 ± 0.035 | 0.49 ± 0.010 |
>
>
>
> We found that β=0 provides competitive mean accuracy with the low variance, confirming our default choice. Non-zero β values **can** yield marginal improvements in ATE or LPIPS, but variability increases and our coarse search did not pinpoint a universally optimal value.
>
> Given the minimal gains versus added complexity of tuning β, we **retain β=0 in our current formulation**. In future work, we intend to explore learning β as a per-scene or per-dataset parameter to enable fine-grained angular attenuation without extensive manual search.
>
> ## Experiment Robustness
>
> We have included updated tables with averages and standard deviations for  5 runs per trial. Due to lack of time and resources during the rebuttal period, we only included evaluations for EndoGSLAM on C3VD, and MonoGS on Self-Capture. We will include all 5 trials, for all experiments, in the final version.
>
> Table 1. Quantitative Evaluation on the C3VD under dynamic lighting, comparing EndoGSLAM with standard Photo-BA versus NFL-BA.
> | depth  | BA    | ATE_T | ATE_R| Chamfer |
> | ------ | ----- | ----- | ---- | ----- |
> | Oracle | Photo | **1.92 ± 0.02** | 1.81 ± 0.02 | **0.85 ±  0.01** |
> |        | NFL   | 2.04 ± 0.06 | **1.12 ± 0.09** | 1.00 ±  0.01 |
> | PPS    | Photo | 3.12 ± 0.14 | 1.76 ± 0.07 | 1.22 ± 0.01 |
> |        | NFL   | **2.56 ± 0.27** | **1.23 ± 0.03** | **1.19 ± 0.01** |
> | DA-H   | Photo | 5.76 ± 1.26 | 2.22 ± 0.12 | **2.03 ± 0.38** |
> |        | NFL   | **4.75 ± 1.15** | **1.81 ± 0.31** | 2.12 ± 0.45 |
>
> Table 2: Quantitative results on four self-captured indoor scenes under dynamic lighting, comparing MonoGS with standard Photo-BA versus NFL-BA
>
> | Scene   | BA    |          ATE\_T |           LPIPS |
> | ------- | ----- | --------------: | --------------: |
> | Guitars | Photo |     0.29 ± 0.02 |     0.39 ± 0.02 |
> |         | NFL   | **0.17 ± 0.01** | **0.36 ± 0.02** |
> | Porch   | Photo |     0.53 ± 0.04 |     0.51 ± 0.02 |
> |         | NFL   | **0.33 ± 0.16** | **0.50 ± 0.03** |
> | Pool    | Photo |     0.46 ± 0.04 |     0.47 ± 0.01 |
> |         | NFL   | **0.32 ± 0.04** | **0.46 ± 0.02** |
> | Stairs  | Photo |     0.37 ± 0.04 |     0.43 ± 0.03 |
> |         | NFL   | **0.22 ± 0.03** | **0.34 ± 0.02** |
>
> ### Full MonoGS Evaluation
>
> As suggested, we evaluated MonoGS with RGB-only input and compiled all MonoGS experiments below. Overall, NFL-BA yields **substantial reductions in both translational and rotational errors across all depth initialization modes**. Additionally, map quality consistently improves, demonstrating that our lighting-aware loss provides complementary information to geometric depth cues, enhancing SLAM robustness even when low-quality depth is available.
>
> Table 3: Quantitative results on the C3VD dataset with 4 depth inputs including RGB-only. Replacing Photometric BA with NFL-BA significantly improves tracking and mapping **regardless of input depth guidance.**
> | Method | Depth Init | BA      | ATE\_T (mm) | ATE\_R (°) | Chamfer (mm) |    LPIPS |
> | ------ | ---------- | ------- | ----------: | ---------: | -----------: | -------: |
> | MonoGS | No depth   | Photo   |        2.85 |       1.85 |         3.95 |     0.61 |
> |        |            | **NFL** |    **1.36** |   **0.78** |     **1.50** | **0.48** |
> | MonoGS | DaHybrid   | Photo   |        4.63 |       1.69 |         1.34 |     0.52 |
> |        |            | **NFL** |    **2.35** |   **1.14** |     **1.13** |     0.52 |
> | MonoGS | PPSNet     | Photo   |        3.48 |       1.70 |         1.59 |     0.56 |
> |        |            | **NFL** |    **2.18** |   **1.65** |     **0.99** |     **0.53** |
> | MonoGS | Oracle     | Photo   |        2.90 |       1.11 |         1.16 |     **0.50** |
> |        |            | **NFL** |    **1.60** |   **1.49** |     **0.79** |     0.51 |

---

> > ### Comment · Reviewer_BMQ8 · 2025-08-05
> > **Response to authors' rebuttal**
> >
> > Q1: Authors acknowledge that there are lighting model limitations which can't be addressed in this work.
> > Q2: Authors addressed the issues I concerned. They experimentally justified that β=0 is an optimal value. More experimental details had been provided in tables.
> > Based on above statements, I will slightly increase my rating as one core issue has been addressed.

---

> > > ### Author Response · Authors · 2025-08-05
> > > **Thank You**
> > >
> > > Dear Reviewer,
> > >
> > > We appreciate the constructive feedback and thank the reviewer for helping to improve the paper.

---

### Official Review · Reviewer_hNJU · 2025-06-29

**Clarity:** 3
**Significance:** 2
**Originality:** 2
**Rating:** 4
**Confidence:** 2

**Summary:**

This paper introduces Near-Field Lighting Bundle Adjustment Loss (NFL-BA), a approach to SLAM that explicitly models dynamic near-field lighting. Traditional SLAM systems typically assume static, distant illumination, which leads to performance degradation in scenarios with co-located light and camera, such as endoscopy, subterranean robotics, and search and rescue in collapsed environments. The dynamic near-field lighting in these situations often causes trobles to traditional SLAM using photometric BA.

NFL-BA addresses this by integrating near-field lighting into the Bundle Adjustment (BA) loss, optimizing surface geometry and camera parameters so that rendered images' shading variations match the relative distance and orientation between the surface and the camera.

Evaluations show that replacing Photometric Bundle Adjustment (Photo-BA) loss with NFL-BA leads to significant improvements in camera tracking and mapping performance. For instance, it improves MonoGS camera tracking by 37% and EndoGS by 14%, achieving state-of-the-art performance on the C3VD colonoscopy dataset.

**Questions:**

Why NFL-BA degrades the performance on EndoGSLAM with GT depth?

**Ethical Concerns:**

["NO or VERY MINOR ethics concerns only"]

**Final Justification:**

The author addressed my concerns and considering on the contribution I would maintain my score as weakly accept.

**Limitations:**

discussed in weakness

**Paper Formatting Concerns:**

no concerns

**Quality:**

3

**Strengths And Weaknesses:**

Strengths:

- integrating near-field lighting into the Bundle Adjustment (BA) loss and improved the SLAM in endoscope scenarios.
- paper is well-written and well-evaluated

Weaknesses:

- while the results are significant, the method is a bit simple
- the performance on EndoGSLAM is not good
- code is not accessible

---

> ### Author Rebuttal · Authors · 2025-07-31
>
> ## Impact of NFL-BA on EndoGSLAM
>
> EndoGSLAM is a **highly specialized SLAM system** tailored for the C3VD dataset through a set of carefully **engineered priors**, such as constant-velocity motion assumptions and brightness-dependent masking, that together deliver outstanding performance in a very specific setting.
>
> **Dataset specialization:** EndoGSLAM is heavily tuned for C3VD and struggles with out‑of‑domain data such as Colon10K (Figure 7), where motion is less uniform and specular reflections are more prevalent. For both test sequences, EndoGSLAM completely fails to produce a continuous structure due to the absence of ground truth depth and broken assumptions
>
> **Engineered for Ground Truth depth:** When ground‑truth depth is available, EndoGSLAM already performs extremely well, leaving minimal room for improvement. When the depths are perfectly aligned, NFL‑BA’s shading‑based optimization contributes little and may occasionally conflict with the built-in priors. However, under noisy depths, EndoGSLAM performance drops sharply, and NFL‑BA provides significant gains by leveraging physically‑based lighting cues to stabilize pose and reconstruction.
>
> In short, the small degradation under GT depth reflects EndoGSLAM’s rigidity and high baseline rather than a fundamental weakness of NFL‑BA. In typical clinical scenarios, NFL‑BA offers a clear benefit.
>
> ## Perceived Simplicity of NFL‑BA
>
> Although NFL-BA is conceptually straightforward, its novelty is in **embedding a physically grounded near-field lighting model directly into a differentiable bundle-adjustment loss**. To our knowledge, no prior Bundle Adjustment (BA) formulation explicitly accounts for dynamic, co-located illumination. This reformulation not only captures essential lighting physics but also **remains stable during optimization.** Moreover, the method’s minimal implementation overhead makes it a **drop-in enhancement** for a wide range of neural SLAM systems without altering their core architectures. Despite its simplicity, NFL-BA delivers state-of-the-art results, reducing MonoGS’s absolute trajectory error by 37% and EndoGS’s by 14% on the challenging C3VD colonoscopy dataset.
>
> ### Code Availability
>
> We understand the importance of reproducibility and will release the full codebase upon acceptance.

---

> > ### Comment · Reviewer_hNJU · 2025-08-07
> >
> > Thanks the authors for addressing my concerns.

---

> > > ### Author Response · Authors · 2025-08-08
> > >
> > > We appreciate the constructive feedback and thank the reviewer for helping to improve the paper.

---

> ### Author Response · Authors · 2025-08-05
>
> Dear reviewer,
>
> Thank you again for your thoughtful and detailed feedback. If you have any further questions or concerns regarding our rebuttal or the paper, we'd be happy to clarify or provide additional context.

---

### Official Review · Reviewer_64q3 · 2025-07-01

**Clarity:** 3
**Significance:** 2
**Originality:** 3
**Rating:** 4
**Confidence:** 4

**Summary:**

Addressing on the assumption of static, distant illumination, the paper proposes a BA loss that explicitly models dynamic near-field lighting (NFL) by incorporating light intensity fall-off based on the relative distance and orientation between the surface and the co-located light and camera. By replacing Photometric Bundle Adjustment loss of SLAM systems with NFL-BA leads to significant improvement in camera tracking on the C3VD colonoscopy dataset.

**Questions:**

Please response to the weakness part.

**Ethical Concerns:**

["NO or VERY MINOR ethics concerns only"]

**Final Justification:**

After reading the rebuttal and considering the comments of other reviewers, I decide to keep the original rating (Borderline accept).

**Limitations:**

Yes

**Paper Formatting Concerns:**

No Paper Formatting Concerns

**Quality:**

3

**Strengths And Weaknesses:**

Strengths

1.A Near-Field Lighting BA Loss that explicitly models near-field lighting as a part of BA loss to enable better performance for scenes captured with dynamic lighting.

2.NFL-BA can be integrated into neural rendering-based SLAM systems with implicit or explicit scene representations.

Weaknesses

1.Implementation for different SLAM systems. In the Supplementary Materials, the paper provides the detials of coefficientsλren andλgeo, which shows different values in different SLAM systems. In my opinion, this may be a limitation for actual applications. Meanwhile，a consequent question is that are the coefficients different for different datasets?

2.Results on colonoscopy dataset. The experimental results focus on the on colonoscopy dataset. Two aspects of explanations can be presented.

(1)The motivation for usage of colonoscopy dataset. For the compared SLAM systems, like MonoGS, EndoGS and NICE-SLAM, they are all tested on the general indoor scenes. Why authors select the colonoscopy dataset as the main testing dataset.

(2)The results on TUM RGB-D and Replica. The compared SLAM systems all report the results on TUM RGB-D and Replic. It is interesting that can the proposed Near-Field Lighting BA improve localization and redenrring performance on these two datasets?

---

> ### Author Rebuttal · Authors · 2025-07-31
>
> ## Choice of Hyperparameters
> We appreciate the reviewer’s concern and would like to clarify that we did **not** tune the hyperparameters {λ_ren, λ_geo} individually for different SLAM frameworks. Instead, we adopted the same values used by the respective base methods. Neural SLAM systems such as MonoGS, EndoGS, and NICE-SLAM inherently use different hyperparameter settings for photometric loss (λ_ren​) and geometric loss (λ_geo​), both in the tracking and mapping stages, due to differences in their underlying 3D scene representations and optimization strategies (e.g., keyframe selection, voxel grid resolution).
>
> Importantly, **we did not modify or tune these values for different datasets either**. The same set of hyperparameters was used consistently across all evaluation sequences and datasets—including C3VD and our in-the-wild captures.
> This design choice reflects our goal of making **NFL-BA a plug-and-play bundle adjustment loss**, compatible with existing neural SLAM pipelines without requiring any additional tuning. By relying on the default settings of each framework, we demonstrate that NFL-BA can be seamlessly integrated and consistently improve performance across diverse settings.
>
> ## NFL-BA on scenes captured with static lighting (TUM RGB-B, Replica)
>
> Near-Field Lighting Effects occur when a light source moves very close to the surface.  In endoscopy, the light source and camera move together within millimeters of the tissue, producing a drastic inverse-square fall-off, deep shadows, and specular highlights. These dynamic lighting phenomena are more extreme in endoscopy than in typical settings. General indoor SLAM datasets (TUM RGB-D, Replica) **assume far-field or static lighting and thus do not exhibit the dynamic illumination** patterns that NFL-BA is designed to correct. Hence, applying NFL-BA on these datasets will not make any sense.
>
> Note that NFL-BA is designed as a plug-and-play Bundle Adjustment loss for any Neural SLAM algorithm, both NeRF & 3DGS-based. This means for any SLAM pipeline, we can easily replace Photometric Bundle Adjustment designed for static lighting with NFL-BA designed for dynamic near-field lighting, without any need to change the SLAM framework.
>
> ## Motivation for using the colonoscopy dataset for benchmarking
>
> Endoscopy provides a **highly challenging near-field lighting scenario**, making it an ideal testbed to demonstrate NFL-BA’s efficacy.
> Endoscopy is a minimally invasive imaging technique where a slender and flexible tube equipped with a light and camera is used to inspect internal organs and tissues of the body via an opening, e.g. mouth or anus. Endoscopy is routinely used for cancer screening, e.g. in colons and lungs; for identifying potential issues in the digestive tract, e.g. ulcers, polyps, hemorrhoids; and respiratory problems, e.g. infections, an abnormal narrowing of the airway, and collapsed lungs, and many others. Robust 3D reconstruction and camera tracking from endoscopy enable various downstream applications: (i) improving 3D visualization during an endoscopy procedure and providing guidance to unsurveyed regions, (ii) automatic quantification of various geometric measures of airway or colon to assist in abnormality detection, and (iii) enabling assisted or automatic navigation of the endoscope without tissue damage.
> Since high-quality publicly available endoscopy datasets for the airway are unavailable, we mainly focused on colonoscopy, since C3VD provides high-quality data with ground-truth.

---

> > ### Comment · Reviewer_64q3 · 2025-08-06
> >
> > Thanks the authors for the detailed response. I will keep my rating positive given most of my concerns are well addressed.

---

> > > ### Author Response · Authors · 2025-08-08
> > >
> > > We appreciate the constructive feedback and thank the reviewer for helping to improve the paper

---

### Official Review · Reviewer_moAc · 2025-07-05

**Clarity:** 3
**Significance:** 2
**Originality:** 3
**Rating:** 4
**Confidence:** 4

**Summary:**

This paper introduces the Near-Field Light Bundle Adjustment (NFL-BA) loss function, a general loss function that can be injected to improve the performance of neural SLAM systems in environments with some scenarios, like dynamic, co-located near-field lighting. Specifically, NFL-BA optimizes the surface geometry and the camera parameters such that the rendered image has shading variations that match the relative distance and orientation between the surface and the camera. The experiment demonstrates the effectiveness of the proposed NFL-BA, particularly in endoscopic procedures and indoor scenes captured with a moving flashlight.

**Questions:**

1. How does the NFL-BA perform on longer test sequences?
2. Can you provide a more detailed analysis of how the improved rendered loss contributes to the overall system when depth input is available?
3. How does NFL-BA compare with simpler baseline methods, such as using only the central part of the image for optimization in scenarios with flashlight usage?

**Ethical Concerns:**

["NO or VERY MINOR ethics concerns only"]

**Final Justification:**

The rebuttal addressed most of my concerns. The new experiments with the center-crop baseline and long sequences further demonstrate the advantages of the proposed method. So I will keep the positive rating.

**Limitations:**

Some limitations are discussed, and I am also curious about the issues listed in Q1 and Q2.

**Quality:**

2

**Strengths And Weaknesses:**

**Strengths:**
* The paper presents a simple yet effective BA loss design in order to address the challenges of dynamic lighting in neural SLAM.
* The proposed NFL-BA can be applied as a plug-in replacement for the existing photometric bundle adjustment in NeRF-based and 3DGS-based methods.
* Several datasets and setups are tested to show the effectiveness of the proposed method.

**Weaknesses:**
* The sequences used for evaluation are relatively short. Longer test sequences are expected to provide a more robust validation of the method.
* From Table 1, with the input depth, the improvement over existing methods is not significant. This raises concerns about how much the improved rendered loss contributes to the overall system when depth input is available. Given that both NeRF-based and 3DGS-based methods are generally not robust without depth input, this issue is crucial for the practical applicability of the proposed method.
* I think there may be a simple baseline comparison. For instance, a more straightforward approach could involve using only the central part of the image during the optimization process for scenarios where a flashlight is used.
* The ATE results in terms of translation and rotation in Table 1 are consistent. More explanation is expected.
* Minor points: 1) It is hard to compare the results in Fig. 7. 2) The SOTA labels in Table 2 are incomplete.

---

> ### Author Rebuttal · Authors · 2025-07-31
>
> ## Evaluation of Center-Crop Baseline
> We thank the reviewer for suggesting the center-crop baseline. We compared MonoGS optimized on only the central 75% and 50% of each frame against full-frame MonoGS + NFL-BA on two sequences, measuring translational ATE (ATE_T), Chamfer distance (CD), and reconstructed point count.
>
> Table A: Center-cropping does improve trajectory error upon the original baseline, but suffers from map sparsity, while full-frame NFL-BA maintains both accuracy and dense reconstructions.
> | Method              | trans\_t3\_a ATE\_T | trans\_t3\_a CD | trans\_t3\_a Points | desc\_t4\_p2 ATE\_T | desc\_t4\_p2 CD | desc\_t4\_p2 Points |
> | ------------------- | ------------------: | --------------: | ------------------: | ------------------: | --------------: | ------------------: |
> | **MonoGS + NFL-BA** |            **0.26** |        **0.60** |          **28,196** |            **0.13** |        **0.67** |           **8,879** |
> | MonoGS (full frame) |                0.31 |            0.76 |               7,095 |                0.25 |            0.76 |               6,398 |
> | MonoGS (75% center) |                0.35 |            1.06 |               5,761 |                **0.13** |            0.97 |               4,329 |
> | MonoGS (50% center) |                0.40 |            2.82 |               1,865 |                0.15 |            1.56 |               2,100 |
>
>
> Although **center-cropping improves camera tracking performance (ATE_t)** of MonoGS and gets close to NFL-BA, it results in **significantly worse reconstruction in terms of quality (CD) and density** (number of points). This is because for camera localization, not all the pixels are essential, and focusing only on central pixels in the image can eliminate the effects of near-field lighting. In contrast, for reconstruction, all pixels in the image matter. This shows the necessity for developing a principled mechanism for handling dynamic near-field lighting, as presented by our work NFL-BA.
>
> ## NFL-BA Performance Variation with Length of Test Sequence
>
> We thank the reviewer for suggesting a thorough evaluation on extended trajectories. C3VD's Seq2 spans  over 4,000 frames with only ground-truth poses, allowing us to measure cumulative drift over time. We note that Seq2 is a screening video that was not utilized in the original paper.
>
> Table B: We evaluate ATE_T (cm) on C3VD Sequence 2 at varying lengths. NFL-BA matches or beats Photo-BA on short sequences and **demonstrates long-term stability**.
>
> | BA         | 500 frames | 1 000 frames | 2 000 frames | 4 000 frames |
> | ---------- | ---------: | -----------: | -----------: | -----------: |
> | Photo  |   2.599 cm |     **1.937 cm** |     8.459 cm |    14.790 cm |
> | **NFL-BA** |   **2.422 cm** |     2.254 cm |     **5.742 cm** |    **11.368 cm** |
>
>
> We observe that NFL-BA matches Photometric BA on short sequences but increasingly outperforms it as trajectory length grows. As a **plug-and-play bundle-adjustment loss**, NFL-BA **inherits the stability of the underlying SLAM system** while further enhancing long-term robustness under dynamic near-field lighting.
>
> ## Performance of NFL-BA with Ground-truth Depth
>
> We acknowledge that NFL-BA’s improvement over existing methods is modest when ground truth depth is available, as reported in Table 1 of the main paper. When **ground truth depth is available, the optimization objective becomes easier.** Hence, **inaccurate modeling of photometric BA loss matters less**. In comparison, when ground truth depth is not present, the optimization is challenging and inaccurate photometric modeling leads to larger error.
>
> In the case of EndoGSLAM, it is a **highly specialized endoscopic SLAM system** with **multiple engineered priors**, including constant‑velocity motion assumptions and brightness‑dependent masking. Unlike methods that rely primarily on photometric cues, EndoGSLAM leverages a diverse set of cues and domain‑specific heuristics. When ground‑truth depth is available, EndoGSLAM already performs extremely well, leaving minimal room for improvement. When the depths are perfectly aligned, NFL‑BA’s shading‑based optimization contributes little and may occasionally conflict with the built-in priors. However, it is heavily tuned for C3VD and **struggles with out‑of‑domain data** such as Colon10K (Figure 7), where motion is less uniform and specular reflections are more prevalent.
>
> Additionally, we would like to address the inconsistencies with translational and rotational ATE. Near-field lighting produces large changes in pixel intensity for small shifts in camera position, making the loss highly sensitive to translational errors. By contrast, rotational misalignments produce subtler shading differences. As a result, our loss yields larger relative gains on translation than on rotation.
>
> ## Evaluation Without Depth Initialization
>
> To quantify how NFL-BA contributes under varying depth guidance, we evaluated MonoGS on the C3VD dataset with four depth initialization modes: no depth (RGB-only), DaHybrid, PPSNet, and Oracle. We compared Photometric BA against NFL-BA and reported both tracking and reconstruction results.
>
> Table C: Quantitative results on the C3VD dataset with 4 depth inputs including RGB-only. Replacing Photometric BA with NFL-BA significantly improves tracking and mapping **regardless of input depth guidance.**
>
> | Method | Depth Init | BA      | ATE\_T (mm) | ATE\_R (°) | Chamfer (mm) |    LPIPS |
> | ------ | ---------- | ------- | ----------: | ---------: | -----------: | -------: |
> | MonoGS | No depth   | Photo   |        2.85 |       1.85 |         3.95 |     0.61 |
> |        |            | **NFL** |    **1.36** |   **0.78** |     **1.50** | **0.48** |
> | MonoGS | DaHybrid   | Photo   |        4.63 |       1.69 |         1.34 |     0.52 |
> |        |            | **NFL** |    **2.35** |   **1.14** |     **1.13** |     0.52 |
> | MonoGS | PPSNet     | Photo   |        3.48 |       1.70 |         1.59 |     0.56 |
> |        |            | **NFL** |    **2.18** |   **1.65** |     **0.99** |     **0.53** |
> | MonoGS | Oracle     | Photo   |        2.90 |       1.11 |         1.16 |     **0.50** |
> |        |            | **NFL** |    **1.60** |   **1.49** |     **0.79** |     0.51 |
>
>
> Overall, NFL-BA yields substantial reductions in both translational and rotational errors across all depth initialization modes. Additionally, map quality consistently improves, demonstrating that our lighting-aware loss provides complementary information to geometric depth cues, enhancing SLAM robustness even when low-quality depth is available.
>
> ### Minor Points
>
> Figure 7 contains results for the colon10k dataset, which contains endoscopy videos captured on real patients in clinical setting, which has no ground truth annotations. To see the source videos or point clouds for the MonoGS results, please check out our supplementary submission site. Additionally, we will ensure that all SOTA labels are complete in Table 2 in the final submission.

---

> ### Author Response · Authors · 2025-08-05
>
> Dear reviewer,
>
> Thank you again for your thoughtful and detailed feedback. If you have any further questions or concerns regarding our rebuttal or the paper, we'd be happy to clarify or provide additional context.

---

> > ### Comment · Reviewer_moAc · 2025-08-06
> >
> > Thank the authors for the feedback. It addressed most of my concerns. The new experiments with the center-crop baseline and long sequences further demonstrate the advantages of the proposed method. I will keep the positive rating.

---

> > > ### Author Response · Authors · 2025-08-08
> > >
> > > We appreciate the constructive feedback and thank the reviewer for helping to improve the paper

---

### Public Comment · ~Andrea_Dunn_Beltran1 · 2026-01-23

We want to acknowledge and correct citation errors in our submission, which were identified by the GPTZero team. These arose when we used an LLM to expand some partial citation cues into full BibTeX entries and did not sufficiently adequately verify the resulting author/venue metadata against canonical sources. This was an oversight on our part. Please note, we intended to cite real and relevant papers, however, errors in the bibliographic details were introduced during the LLM-based completion.

We have submitted an updated version to arXiv (https://arxiv.org/abs/2412.13176) and will reach out to the program chairs of NeurIPS'25 as well. We apologize to the affected authors and to the broader community for the confusion and the additional burden this caused, and we appreciate the opportunity to correct the record.


**Corrections**
* X. Sun, et al. “Lightglue: Feature matching under adverse conditions.” CVPR 2023.
   * Corrected to: Philipp Lindenberger, Paul-Edouard Sarlin, and Marc Pollefeys. “LightGlue: Local Feature Matching at Light Speed.” In ICCV, 2023

* Y. Zhang, at al. “Airslam: Illumination-invariant hybrid slam.” ICCV 2023.
   * Corrected to: Kuan Xu, Yuefan Hao, Shenghai Yuan, Chen Wang, and Lihua Xie. “AirSLAM: An efficient and illumination-robust point-line visual slam system.” IEEE Transactions on Robotics (TRO), 2024.

* Z. Zhu, et al. “Neuralrgb-d: Neural representations for depth estimation and scene mapping.” CVPR 2022.
   * Corrected to: Dejan Azinovi´c, Ricardo Martin-Brualla, Dan B Goldman, Matthias Nießner, and Justus Thies. “Neural rgb-d surface reconstruction.” In Proceedings of the IEEE/CVF Conference on Computer Vision and Pattern Recognition (CVPR), pages 6290–6301, June 2022

---

### Note · Authors · 2025-08-14

We thank the reviewers once again for their thoughtful feedback and we are pleased to note that **all reviews reflect positive ratings** before and after rebuttal. The reviewers **appreciated the novelty, simplicity, and plug-and-play applicability of NFL-BA**, along with its consistent **improvements in camera tracking and mapping under dynamic near-field lighting** for both endoscopy and hand-held camera recordings.


**Rebuttal Period**
*  **Evaluation on longer sequences.** We tested C3VD Seq2 at different frame amounts (500, 1k, 2k, 4k) and demonstrated that NFL-BA matches Photometric BA on short sequences and increasingly outperforms it as sequence length grows.
*  **Additional Baseline: MonoGS with center crop.** We also implemented the suggested center-crop baseline, showing that while it improves trajectory error, it produces sparser and lower-quality reconstructions than NFL-BA.
*  **Comparison with NFL-BA without depth initialization.** We clarified that NFL-BA yields its largest gains when depth is absent or noisy, complementing geometric cues. With perfect ground-truth depth, improvements are naturally modest, and minor degradations reflect strong, highly tuned priors rather than a weakness of NFL-BA.
*  **Robustness with optimization hyper-paramaters.**  NFL-BA can be used as a plug-and-play style Bundle Adjustment loss without any data or SLAM specific tuning, and the same hyperparameters were used across datasets.
*  **Acknowledgment of limitations in modeling complex lighting effects.** We acknowledge the diffuse-only model does not capture specularities, subsurface scattering, or inter-reflections. Our specularity masking mitigates this. We justified β=0 for endoscopic LEDs and showed via ablation that it remains competitive for smartphone flashlights.

**Reviewer Updates**
*  **Reviewer moAc**: maintained positive rating
*  **Reviewer 64q3**: maintained positive rating
*  **Reviewer hNJU**: maintained positive rating
*  **Reviewer BMQ8**: **increased** already positive rating

In summary, **simplicity and effectiveness** of the NFL-BA loss function make it an attractive **plug-and-play solution** for improving the performance of existing neural SLAM systems without requiring extensive modifications to their core architectures. NFL-BA is extremely effective when scene is captured under dynamic lighting conditions, with applications in **endoscopy, subterranean robotics, and search-and-rescue operations in collapsed environments**.

---

### Decision · Program_Chairs · 2025-09-17

**Decision:**

Accept (poster)

**Comment:**

This paper addresses the performance degradation of SLAM under dynamic near-field lighting by proposing a new bundle adjustment loss, NFL-BA. As a plug-and-play replacement for existing photometric BA, NFL-BA incorporates a physics-based light attenuation model and can be broadly applied to both NeRF-based and 3DGS-based SLAM. Experiments on endoscopic datasets (C3VD, Colon10K) and scenes captured with smartphone flashlights demonstrate improvements in both camera tracking and mapping accuracy.

While some reviewers noted the simplicity of the method as a potential weakness, this same property was also recognized as a strength, since NFL-BA can enhance existing SLAM systems with minimal modifications. Concerns raised included: (1) the lack of validation on standard SLAM benchmarks, (2) the assumption of a purely diffuse reflection model, which does not account for specularities or subsurface scattering, and (3) only modest improvements in scenarios with strong priors or reliable depth input. However, the authors addressed these points in the rebuttal by providing additional experiments, including long-sequence evaluations, comparisons with a center-crop baseline, and analyses across different depth initialization settings.

Overall, the reviewers maintained or strengthened their positive assessments, confirming both the practicality and the impact of the proposed approach.